# Biomarkers Predictive of Metabolic Syndrome and Cardiovascular Disease in Childhood Cancer Survivors

**DOI:** 10.3390/jpm12060880

**Published:** 2022-05-27

**Authors:** Alberto Romano, Ester Del Vescovo, Serena Rivetti, Silvia Triarico, Giorgio Attinà, Stefano Mastrangelo, Palma Maurizi, Antonio Ruggiero

**Affiliations:** Pediatric Oncology Unit, Fondazione Policlinico Universitario A. Gemelli IRCCS, Università Cattolica Sacro Cuore, 00168 Rome, Italy; alberto.romano@guest.policlinicogemelli.it (A.R.); esterdelvescovo@gmail.com (E.D.V.); serena.rivetti@gmail.com (S.R.); silvia.triarico@guest.policlinicogemelli.it (S.T.); giorgio.attina@policlinicogemelli.it (G.A.); stefano.mastrangelo@unicatt.it (S.M.); palma.maurizi@unicatt.it (P.M.)

**Keywords:** cancer survivors, metabolic syndrome, cardiovascular disease, childhood cancer, chemotherapy toxicity, radiotherapy toxicity, obesity

## Abstract

The improvement in childhood cancer treatments resulted in a marked improvement in the survival of pediatric cancer patients. However, as survival increased, it was also possible to observe the long-term side effects of cancer therapies. Among these, metabolic syndrome is one of the most frequent long-term side effects, and causes high mortality and morbidity. Consequently, it is necessary to identify strategies that allow for early diagnosis. In this review, the pathogenetic mechanisms of metabolic syndrome and the potential new biomarkers that can facilitate its diagnosis in survivors of pediatric tumors are analyzed.

## 1. Introduction

In recent years, the improvement in childhood cancer treatments and the adoption of international cooperative treatment regimens, which permit the combination of surgery, radiotherapy and chemotherapy, resulted in a marked increase in survival [1,2]. In parallel with the increase in survival, increased toxicity was observed, and treatment-related long-term side effects were noted [3,4]. Chemotherapy, high-dose steroid therapy and radiotherapy cause long-term toxic effects on numerous organs, including the kidneys, heart, endocrine system, and ear [5,6,7]. These treatments are also the cause of chronic inflammation and metabolic alterations, resulting in the onset of metabolic syndrome (MetS) and consequent increase in cardiovascular risk [8]. Cardiovascular-related death is seven times more frequent in childhood cancer survivors (CCS) than in the general population. It is the cause of a quarter of all deaths within 45 years of cancer diagnosis [9]. Such a high incidence explains the need to carry out careful monitoring of CCS to diagnose the onset of MetS early and implement measures aimed at reducing the risk of cardiovascular-related death. This review analyzes the pathogenetic mechanisms underlying the onset of MetS and cardiovascular diseases in CCS and the new biomarkers that allow them to be diagnosed early.

### Research Methods

This research aimed to write an integrative review to summarize the knowledge currently available on the pathogenesis of MetS and cardiovascular diseases in CCS and their new diagnostic biomarkers. To reach this goal, we searched for papers dedicated to biomarkers of MetS and cardiovascular diseases in CCS, and performed a Pubmed-based retrieval of articles using the search terms “Metabolic syndrome”, “Cardiovascular risk” and “biomarkers” matched with “cancer survivors” and “biomarkers”. After the original search, we used filters to select articles available in the English language and articles with available full texts. This research retrieved 220 articles. Two operators set the 220 articles according to the adherence of the title and abstract to the topic.

The literature review was later expanded to search for single biomarkers matched to “cancer survivors” filtered to select articles available in the English language and articles with available full texts. This research retrieved 47 articles that the same two operators analyzed.

A total of 150 papers were obtained and included in the review at the end of this search.

Figure 1 summarizes the research methods.

## 2. Pathogenesis of Metabolic Syndrome and Cardiovascular Disease in CCS

MetS is a group of symptoms that includes obesity, impaired glucose tolerance and dyslipidemia. It is also characterized by inflammatory and prothrombotic states [10,11]. Many definitions describe MetS, but the latest consensus of the International Diabetes Federation, the National Heart, Lung and Blood Institute and the American Heart Association, in 2009, defined diagnostic criteria to diagnose MetS for adult patients [12,13]. At least three of the following are required to diagnose MetS:Raised Waist Circumference (population- and country-specific definitions)Fasting Plasma Glucose Concentration ≥ 100 mg/dL or on diabetes treatmentBlood Pressure ≥ 130/85 mmHg, or on antihypertensive treatmentTriglycerides ≥ 150 mg/dL or on treatmentHigh-density lipoprotein cholesterol < 40 mg/dL in men and < 50 mg/dL in women, or on treatment.

On the other hand, pediatric patients have no univocal guidelines for the diagnosis of MetS. The definitions available now share the following criteria: central obesity, hypertension, hypertriglyceridemia, low HDL, and impaired glucose [14].

CCS have an increased risk of developing MetS compared to their siblings, and a 10 times higher risk of developing cardiovascular disease [15]. The pathogenesis of MetS in CCS is not well known. Still, many studies underline the role of low-grade chronic inflammation due to cytokine activation released from abdominal fat determined by the direct action of treatments on organs and the cardiovascular system. Radiotherapy, chemotherapy and prolonged, high-dose steroid therapy can interfere with metabolic processes, facilitating the onset of MetS [16].

### 2.1. Radiotherapy

Cranial radiotherapy causes hypothalamus/hypophysis axis dysfunction, which is associated with an increased android/ginoide fat ratio and consequential central fat accumulation, which is responsible for releasing inflammatory molecules. High dose cranial radiation (>30 Gy to the hypothalamic-pituitary axis) determines leptin resistance on hypothalamic receptors and its increased circulation levels [17]. Leptin is an adipokine produced by adipocytes and its receptors are predominantly expressed in hypothalamic nuclei and the arcuate nucleus. After radiotherapy, this structure can be damaged with the onset of leptin resistance and leptin overproduction by fat tissue [18]. The lack of leptin action on the hypothalamus causes an increase in adipose tissue, confirmed by the finding that high circulating levels of leptin are strongly associated with BMI percentiles for age, sex, and visceral adiposity. The high level of leptin leads to glucose intolerance and insulin resistance [19]. As a consequence of leptin resistance, CCS exposed to cranial irradiation have a higher BMI, fat mass, and central adiposity. Moreover, CCS exposed to cranial radiation develop growth hormone (GH) deficiency, which is associated with elevated fasting insulin, abdominal obesity, and dyslipidemia, independent of radiation dose [20]. There is no significant difference in BMI and trunk fat between patients who received 0–20 Gy and those who received >20 Gy cranial irradiation [21]. In addition, GH substitution in patients with radiation-induced deficiency and a dysregulated hypothalamic-pituitary axis worsens insulin resistance. GH substitution leads to elevated circadian GH levels, enhancing lipid oxidation and free fatty acid production [22]. In CCS of leukemia, it is also demonstrated that the risk of hyperglycemia and insulin resistance is correlated with cranial radiation, which persists after correcting the data for BMI [23].

Radiotherapy causes an increase in the incidence of MetS in CCS even when it is performed in other parts of the body. CCS who received abdomen or chest radiation and steroid therapy have higher central systolic and diastolic blood pressure [24]. The pathogenesis is probably associated with direct vascular injury and fibrosis. In one study, patients exposed to radiotherapy but not to cardiotoxic chemotherapy had decreased left ventricle wall thickness and wall mass similar to those who received anthracyclines [25]. This mechanism was confirmed by a report that demonstrates the strong association between abdomen and liver radiation (>15 Gy) and portal hypertension in CCS of Wilms Tumor [26]. Patients exposed to abdomen radiation have the highest risk of developing type 2 diabetes and insulin intolerance as a result of adipose damage after radiation, which causes cytokine release and chronic low-grade inflammation. The exact mechanism of irradiation-induced damage is associated with mitochondrial injury-inducing hyperlipidemia and fat storage dysfunction [27]. Moreover, in patients with lymphoma treated with abdominal irradiation, the pathogenesis of diabetes is related to radiation damage in survivors that received >10 Gy to the tail of the pancreas, resulting in pancreatic insufficiency [28].

### 2.2. Steroid Therapy

Steroid therapy is implicated in the pathogenesis of MetS as it causes a higher risk of developing type 2 diabetes mellitus and glucose intolerance with significant central fat accumulation, cytokine release and chronic inflammation. Chronic inflammation is responsible for activating ROS and reactive nitrogen species (RNS), causing DNA damage. This process leads to vital organ failure with specific consequences: liver steatosis; premature atherosclerosis, thrombosis and myocardial infarction; osteoporosis and osteopenia; neurocognitive alteration and neuronal tissue damage [29].

Prolonged steroid-induced hyperglycemia associated with a sedentary lifestyle and irregular food intake causes permanent diabetes in CCS. The pathophysiology involves different mechanisms: increased insulin resistance, increased gluconeogenesis, and decreased insulin production. Insulin resistance is caused by increasing hepatic gluconeogenesis by activating genes coding for phosphoenol-pyruvate carboxykinase and glucose-6-phosphatase [30]. Corticosteroids also increase the transport of metabolites across the mitochondrial membranes, facilitating gluconeogenesis. Moreover, it inhibits peripheral use of glucose, leading to lipid accumulation in skeletal muscles and increasing free fatty acid delivery. This mechanism is associated with corticosteroid inhibition of GLUT4 translocation to the cell surface in response to insulin production [31]. Free fatty acids enhance the inhibition of insulin-dependent glucose uptake by peripherical tissues. In the liver, the presence of free fatty acids leads to the production of glucose, triglycerides and apoB, which are atherogenic. In particular, serum apoB is a strong predictor of cardiovascular risk [32]. Production and secretion of insulin from pancreatic beta cells is influenced by dose, time of exposure, and administration of corticosteroid treatment. Intravenous infusion or oral administration at high doses leads to acute inhibition of insulin secretion. Moreover, blood glucose variability during the day also depends on the type of corticosteroid formulation [33].

### 2.3. Chemotherapy

Chemotherapy drugs are involved in the genesis of MetS in CCS by their direct and indirect actions. It is not easy to understand the mechanism through which single chemotherapeutic agents act in determining MetS, as they are frequently administered together. However, it is known how some chemotherapeutic agents cause damage to the cardiovascular system. For example, anthracyclines cause cardiovascular damage and hypertension due to anthracycline-related cardiotoxicity, which causes left ventricular pathological remodeling, fibrosis, and afterload abnormalities [34]. Pathophysiology of doxorubicin-induced cardiomyocyte atrophy and death is related to p53 expression. Doxorubicin induces p53 which is necessary for the inactivation of the mammalian target of rapamycin (mTOR), which in turn is essential for protein synthesis. This leads to myocyte atrophy and reduction in heart weight [35].

CCS treated with platinum and steroids were strongly at risk of developing insulin resistance and cardiovascular risk [28,36]. The pathogenesis is associated with the direct exposure of endothelial cells to platinum, which leads to endothelial cell release of IL-1, IL-6, IL-8 and GM-CSF. Moreover, IL-1 is able to induce superoxide dismutase (SOD) from mitochondria which catalyze the conversion of O2 to H2O2, with consequent endothelial damage [37,38].

Patients affected by Acute Lymphoblastic Leukemia (ALL) receive L-asparaginase during induction therapy associated with corticosteroids. L-asparaginase can directly inhibit insulin biosynthesis, causing impaired intracellular signaling, reducing and modifying insulin receptors, and indirectly reducing insulin production via induction of pancreatitis and beta cells destruction. These mechanisms lead to a systemic insulinopenic state and hyperglucagonemia, enhanced by beta cells cytotoxicity and inflammation [39,40]. Some additional risk factors such as age and genetic conditions (e.g., Down Syndrome) can enhance prednisone-asparaginase induced hyperglycemia in ALL patients. For example, studies demonstrate that ALL patients older than 10 years of age have the highest risk of developing MetS. The pathogenesis is probably related to sex hormone excretion during puberty that can enhance glucose intolerance and insulin resistance [41].

Moreover, all chemotherapy drugs activate inflammatory processes with the accumulation of senescent cells and the increasing of reactive oxygen species facilitates the onset of MetS [27,42]. Chemotherapy causes mucositis that alters normal intestinal flora, disturbs the microbiome, and causes febrile neutropenia with the consequent need to receive broad-spectrum antibiotics [43]. The gut environment can interact with the host through the presentation of various ligands that activate catalytic pathways for the metabolism of complex carbohydrates that produce short-chain fatty acids, anti-inflammatory and anti-proliferative lipids. These molecules modulate immune homeostasis in the gastrointestinal tract and mucous surfaces. Antibiotic- and chemotherapy-induced alterations in the gut microbiome contribute to anti-inflammatory dysfunction and increased cytokine production [44]. This mechanism influences the appearance of MetS.

## 3. Biomarkers Predictive of Metabolic Syndrome and Cardiovascular Disease in CCS

The definition of MetS includes biomarkers such as an increase in triglycerides, a reduction in HDL, and impaired glucose, which is defined as a fasting blood glucose of ≥100 and <126 mg/dL, or blood glucose ≥140 and <200 mg/dL at the 2 h mark of the oral glucose tolerance test [45]. However, the alteration of these biomarkers occurs when the condition is already in place; for this reason, it is necessary to identify biomarkers capable of predicting the manifestations related to MetS in advance in order to implement measures to avoid its appearance.

Based on the pathogenetic mechanisms that lead to the onset of MetS and consequent increase in cardiovascular risk, in recent years, biomarkers capable of predicting the onset of Mets were identified in CCS [19]. Among these are the adipokines adiponectin and leptin, uric acid, the inflammatory markers high sensitivity C-reactive protein (hsCRP), Tumor Necrosis Factor-alpha (TNF-α), interleukin 1 (IL-1) and interleukin 6 (IL-6), and the lipid markers apolipoprotein B (apoB) and lipoprotein(a) (Lp(a)) [19].

The mechanisms that lead to an increase/reduction in these biomarkers, the laboratory systems with which to carry out the measurements, and the reference values are analyzed below and summarized in Figure 2 and Table 1.

### 3.1. Adiponectin

Adiponectin is a protein of 244 amino acids produced by adipocytes and, to a small extent, by cardiac and skeletal myocytes; it is secreted into the bloodstream in three different forms: a trimer, a hexamer, and a high molecular weight multimer [46]. The production and secretion of adiponectin is favored by physical exercise and healthy diet, and is different in the two sexes with greater production in the female sex due to the action of estrogen on adipose tissue [47]. Once released into the blood, adiponectin binds to transmembrane receptors called AdipoR1, expressed in skeletal muscles, and AdipoR2, expressed by hepatocytes, and acts by modulating numerous metabolic processes [46]. At the level of skeletal myocytes, adiponectin increases insulin sensitivity, while in the liver, it up-regulates glucose transport, down-regulates gluconeogenesis and activates the oxidation of fatty acids. Adiponectin also increases insulin sensitivity in the liver, and acts directly on pancreatic cells by increasing insulin secretion [48]. Furthermore, adiponectin plays a role in the modulation of inflammatory processes in macrophages, endothelial tissue, muscles and epithelial cells by preventing the production of reactive oxidative species and inhibiting the secretion of hs-CRP. Through these processes, adiponectin acts in a protective way against inflammatory diseases such as atherosclerosis and MetS [49,50]. In fact, adiponectin inversely correlates with intimal thickness [32,51] and with adiposity and proinflammatory cytokines; low values of adiponectin, especially of the high molecular weight form, are associated with an increased risk of developing MetS [52]. It was shown that adiponectin correlates inversely with adiposity in survivors of brain tumors [32], with the antero-posterior diameter of infrarenal abdominal aorta in survivors of leukemia [51], and with the appearance of MetS in pediatric survivors of lymphoma and allogeneic hematopoietic stem cell transplantation [23,53].

The measurement of adiponectin can be carried out using the enzyme-linked immunosorbent assay (ELISA) technique [51]; numerous commercial kits are also available. Erhardt et al. established age- and sex-specific reference values for serum adiponectin in normal-weight 3.0–8.9 year old European children [54], and Lausten-Thomsen et al. developed reference levels for total serum adiponectin in children and adolescents aged 6–18 years [55]. It is usually expressed in μg/mL.

### 3.2. Leptin

Leptin is a 146 amino acid protein encoded by the ob gene and released from adipose tissue into the blood in quantities directly proportional to the amount of adipose tissue [56]. Leptin acts to bind a specific receptor present on neuronal, hepatic, pancreatic, cardiac, and perivascular intestinal tissue. At the brain level, leptin has as its main sites of action the solitary tract and the ventral tegmental area in the brain stem, where it reduces appetite by stimulating neurons secreting proopiomelanocortin and inhibiting the orexigenic agouti-related protein/neuropeptide Y-containing (AgRP/NPY) neurons [57]. It also regulates the axes of the thyroid gland, gonads, adrenocorticotropic hormone and cortisol growth hormone, and changes in cognition, emotions, memory, and the entire brain structure [58,59]. High quantities of leptin are produced in the case of excess adipose tissue, and this determines the inhibition of the sense of hunger and consequent reduction in food intake. Leptin deficiency or resistance is associated with dysregulation of cytokine production, increased susceptibility to infections, autoimmune disorders, malnutrition, and inflammatory responses [57]. The absence of leptin is the cause of a pathological condition characterized by severe obesity, hyperinsulinemia and dyslipidemia [60,61,62]. Leptin plasmatic value is influenced by sex and gender and is greater in females than in males in both children and adults [63]. In adults, leptin is positively correlated with fasting insulin concentrations [64] and is a predictor of glucose intolerance, insulin resistance and MetS regardless of underlying obesity [65]. Furthermore, elevated leptin levels were found to be a significant predictor of cardiovascular-related death and hypertension [66,67]. In children, leptin correlates with the onset of MetS. Madeira et al. demonstrated that in prepubertal children, leptin levels above 13.4 ng/dL were significantly associated with MetS and that, for every 1 ng/dL increase in leptin levels, the odds of MetS increase by 3% [68]. In CCS of brain tumors, plasma leptin values were higher than in healthy subjects and correlate with central fat indicators such as waist-to-hip ratio and waist-to-height ratio [36]. Additionally, in CCS of leukemia and lymphoma and those who survived to hematopoietic stem cell transplantation, leptin levels were demonstrated to be associated with each of the components of MetS [23,69]. The measurement of leptin can be carried out with the ELISA technique, and numerous commercial kits are available. According to Gijón-Conde et al., leptin values that identify cardiometabolic abnormality are 23.75 ng/mL in women and 6.45 ng/mL in men [70]. There is no strong evidence of normal pediatric leptin values [71]. Savino et al. reported that in a group of 317 infants, the median leptin concentration was 2.81 ng/mL in infants younger than 6 months of age, 1.44 ng/mL in infants between 6–12 months of age and 1.77 ng/mL in infants between 12–18 months of age; in addition, they obtained leptin reference values based on age using estimates of the lower and upper percentiles and revealed no gender difference in leptin concentration in early infancy [72]. Instead, Erhardt et al. established age- and sex-specific reference values for serum leptin in normal-weight 3.0–8.9 year old European children [54]. The most frequently used unit of measurement is ng/mL.

### 3.3. Uric Acid

Uric acid is the product of purine metabolism, and it is eliminated from the body in part via uric acid transporters present in the kidney and intestinal tract [73], and the remainder is eliminated via the substrate of hypoxanthine-guanine phosphoribosyltransferase, which recycles purines [74]. One of the causes of an increase in uric acid serum levels is the intake of foods and beverages rich in purines such as meat, seafood, alcohol, and beverages and foods containing high amounts of sugar, such as fructose. Excessive intake of fructose causes the consumption of large quantities of ATP with the production of ADP and AMP, which are metabolized, resulting in the production of uric acid [75,76]. Another cause of hyperuricemia is insulin resistance and high plasma insulin concentrations. In studies carried out on mice, insulin acts at the renal level favoring the expression of the uric acid reabsorption system and decreasing the expression of a major urate secretory transporter [77]. In humans, it was widely demonstrated that insulin values correlate with uric acid values and reduce urinary excretion of uric acid, although the mechanism underlying this phenomenon is not fully known [78,79,80]. The excessive concentration of uric acid in the cells causes an increase in the activity of xanthine oxidase and causes damage to the mitochondria with a consequent increase in the production of reactive oxygen species [81]. Furthermore, uric acid promotes the production of inflammatory cytokines. The production of reactive oxygen species and the activation of the inflammatory system stimulates the well-known process of atherosclerosis, increasing the risk of cardiovascular diseases in subjects with hyperuricemia [81]. Oxidative stress caused by uric acid, in turn, determines an increase in insulin resistance, fatty liver, and dyslipidemia resulting in a vicious circle that causes MetS and an increase in cardiovascular risk [82]. Pluimakers et al. observed that in CCS of abdominal cancer subjected to radiotherapy, uric acid is a predictive indicator of MetS and allows the early identification of subjects at risk of developing it [18]. The same evidence was also obtained in the CCS of allogeneic hemato-poietic stem cell transplantation and leukemia [53,83]. The determination of uric acid in serum can be accomplished using numerous approaches, such as capillary electrophoresis, fluorometry, chromatography, electrochemical methods, chemiluminescence, and colorimetry. The colorimetric method is the most widely used due to its ease of use, high analysis speed, and high sensitivity [84]. In healthy adults, uric acid must be less than 6.6 mg/dL or 360 µmol/L [85]. Uric acid values are lower in pediatric patients and should be compared with age- and gender-adjusted percentiles [86]. It is usually expressed in mg/dL or µmol/L.

### 3.4. Hs-RCP

Hs-CRP is a pentameric protein synthesized by the liver, whose production is induced by IL-6 during the acute phase of the inflammatory/infectious process. Hs-CRP carries out proinflammatory and also anti-inflammatory activities [87]. It recognizes and promotes the removal of foreign pathogens and damaged cells by binding to phosphocholine, phospholipids, histone, chromatin and fibronectin. Hs-CRP also activates the classical complement pathway and phagocytic cells via immunoglobulin Fc receptors, accelerating the removal of cell debris and damaged or apoptotic cells and foreign pathogens. In some cases, hs-CRP can amplify tissue damage caused by pathogens or autoimmune diseases by activating the complement system, and, therefore, inflammatory cytokines [88,89]. It is also involved in chronic infectious and non-infectious inflammatory processes, and sometimes mild elevations in hs-CRP can be seen without any systemic or inflammatory disease, such as in obesity, insomnia, depression, etc. [87]. Insulin resistance, atherosclerosis, and cardiovascular disease are associated with chronic low levels of systemic inflammation and hs-CRP levels in adults and children [90]. In CCS, exposure to oncogenic insults (chemo- and radiotherapy) induce a persistent activation and recruitment of immune cells, such as lymphocytes and macrophages, determining the production of pro-inflammatory molecules and amplifying the inflammatory response leading to inflammation, the accumulation of senescent cells, and the increasing of reactive oxygen species and DNA mutations [42,91]. This chronic low-grade inflammation facilitates the onset of MetS in CCS and the general population [92,93]. The close relationship between inflammation and MetS in CCS is evidenced by numerous studies that show correlations between the values of hs-CRP with each of the components of MetS [19,94,95]. The measurement hs-CRP can be performed using immunological tests and laser nephelometry with results reported in mg/dL or mg/L. When used for cardiac risk stratification, hs-CRP levels below 1 mg/L are considered low risk. Levels between 1 mg/L and 3 mg/L are considered moderate risk, and a level above 3 mg/L is deemed to be at high risk for the development of cardiovascular disease [96,97].

### 3.5. TNF-α

TNF-α is a cytokine produced by immune and non-immune cells and acts by binding to the receptors of TNFR1 (constitutively and ubiquitously expressed) and TNFR2, which is expressed on lymphocytes and endothelial cells, but can be induced in response to TNFR1 activation and signaling [98]. It is involved in innate and adaptive immunity and in the normal function of immune cells. Sustained and elevated TNF-α production is associated with pathogenic inflammatory disease states, including infection-related sepsis and chronic autoimmune diseases [99]. However, it was seen that TNF-α is abundantly produced in the adipose tissue in obese subjects and that it has a role in mediating insulin resistance [100] and regulating metabolism. TNF-α stimulates hepatic lipid synthesis, and fatty lipolysis in adipose tissue promotes cholesterol and apolipoprotein biosynthesis while decreasing cholesterol catabolism and excretion as bile acids [101]. In addition, TNF-α promotes hypertension, inducing vascular insulin resistance, reducing vasodilation, increasing intravascular fluid and vasoconstriction, and promoting sympathetic overactivity [102]. Being involved in such a large number of processes, TNF-α is one of the fundamental molecules in the pathogenesis of MetS.

In CCS of leukemia, TNF-α was observed to be higher than in controls [43]. Although the crucial role of TNF in the pathogenesis of MetS is evident, there is currently little evidence regarding the usefulness of the assay in CCS [19]. The measurement of TNF-α can be carried out with the ELISA technique, and numerous commercial kits are available. TNF-α values are higher in children than in adults; however, no well-defined reference values for age are available [103]. It is usually expressed in pg/mL.

### 3.6. IL-1

IL-1 is a cytokine with a wide range of biological functions, including acting as a leukocytic pyrogen, a mediator of fever and a leukocytic endogenous mediator, and an inducer of several components of the acute-phase response lymphocyte-activating factor [104,105,106]. There are two different isoforms of IL-1, IL-1α and IL-1β, which perform the same biological functions [107]. IL-1α and IL-1β are produced in a wide variety of cells, especially in macrophages in lymphoid organs. In non-lymphoid organs, IL-1α and IL-1β are expressed in tissue macrophages in the lung, digestive tract, liver, glomeruli, and various specific cell types, including neutrophils, epithelial and endothelial cells, lymphocytes, smooth muscle cells and fibroblasts [108,109]. In addition to intervening in the modulation of inflammatory processes and innate immunity, IL-1 plays a role in the pathogenesis of MetS. High concentrations of glucose and low-density lipoproteins that are produced in the course of MetS are able to favor the production of IL-1 [110,111], and IL-1α and IL-1β gene polymorphisms were reported to be associated with central obesity and MetS [112]. Furthermore IL-1, in particular IL-1β, was observed to have an insulin resistance action; as identified by Spranger et al. in a group of 27,500 subjects, increased plasma IL-1β, as well as IL-6 levels, increased the risk of developing type 2 diabetes within a 2.3 year period [113]. Necrotic adipocytes release “warning signals” capable of activating the production of IL-1α, which recruits innate immune cells into adipose tissue. Since adipocyte death is increased in adipose tissue during obesity, IL-1α plays a pivotal role in the initiation of adipose tissue inflammation during obesity by promoting the chronic inflammation typical of MetS [114,115]. In adults, IL-1 was shown to be highly expressed in several types of tumors, including breast, colon, head and neck, lung, and pancreas tumors, and melanomas [116]. In children with leukemia and a solid tumor, high concentrations of IL-1 were identified [117,118]. However, there is little evidence of the role of IL-1 in the pathogenesis of MetS in CCS [19]. The ELISA technique can be used for the assay of IL-1, and several commercial kits are available. Berdat et al. identified the reference values in relation to the age of the patients [119]. It is usually expressed in pg/mL.

### 3.7. IL-6

IL-6 is a 212 amino acids cytokine involved in immune responses and inflammation, hematopoiesis, bone metabolism, embryonic development, and other fundamental processes [120]. It acts on hepatocytes inducing the synthesis of acute-phase proteins such as hs-CRP, serum amyloid A, fibrinogen, and hepcidin, whereas it inhibits albumin production [121]. IL-6 plays an important role in acquired immune response by stimulating antibody production and effector T-cell development. IL-6 stimulates megakaryocytopoiesis in the bone marrow and acts as an osteoclast differentiation modulator [122]. In addition to these functions, IL-6 plays an important role in various metabolic processes as autocrine and/or paracrine actions of adipocyte function [123] and is closely linked to MetS favoring the onset of insulin resistance, elevated glucose production in the liver, inhibition of the insulin-mediated glucose uptake in skeletal muscle, and facilitating the onset of hypertension [124]. Furthermore, the enlargement of adipose tissue in obesity induces mechanical stress and hypoxia in adipocytes, resulting in the release of free fatty acids and inflammatory cytokines such as IL-6 and TNF-α, with the consequent generation of chronic inflammation and amplification of the pathogenetic mechanisms of MetS [125]. It is demonstrated that in adults, IL-6 plays a role in the progression and severity of many forms of cancer [126], and it correlates with poor prognosis in children with neuroblastoma and acute myeloid leukemia [127,128]. Higher IL-6 values were also found in leukemia survivors [43]; however, there is not much evidence for the role of IL-6 in the pathogenesis of MetS in CCS [19]. IL-6 can be assayed using the ELISA technique, as well as several commercial kits. Berdat et al. identified the reference values in relation to the age of the patients [119]. It is usually expressed in pg/mL.

### 3.8. ApoB

Apolipoproteins are a group of proteins involved in transport in the various tissues of lipids, which are not soluble in plasma [129]. Among these, apoB is responsible for the transport of chylomicrons, low-density lipoprotein (LDL), very-low-density lipoprotein (VLDL), intermediate-density lipoprotein (IDL), and lipoprotein(a) [130]. The same gene encodes two types of apoB: apoB100 is synthesized in the liver and is a component of VLDL and LDL; apoB48 is expressed in the intestine and is present in chylomicrons and their remnants [131]. Of the two forms, apoB100 is the one mainly involved in the formation of atherosclerotic plaques. ApoB48 transports chylomicrons from the intestine to the liver. In the liver, free fatty acids generated from chylomicron residues are used to produce triglycerides incorporated into nascent VLDLs. VLDL particles, each containing a single molecule of apoB100, are secreted by the liver into the blood. VLDL particles shrink with the loss of surface components in HDL and are catabolized into IDL by lipoprotein lipase. Then, IDL is converted to LDL. LDL can be oxidatively modified and absorbed by macrophages, which leads to excessive accumulation and the formation of foam cells which are the initial components of atherosclerotic plaques [132]. At least one apoB molecule is present in all atherosclerotic plaques and for this reason it was proposed as a predictive biomarker of cardiovascular events. In fact, recent studies show that apoB has a higher sensitivity and specificity than LDL in predicting cardiovascular events, such as myocardial infarction in both men and women, independent of age [133]. Patients with high levels of apoB have a higher BMI, waist circumference, systolic blood pressure, fasting insulin and C-reactive protein, which are all components of MetS [134], and epidemiological studies show that apoB predicts the development of type 2 diabetes as much as 3–10 years in advance of clinical onset [135].

Broberg et al. demonstrated high values of apoB in CCS subjected to a high dose of anthracycline [136], and the same observation was shown in CCS of leukemia [137].

The ELISA can measure apoB, but this technique may be expensive and time-consuming, and its accuracy may vary [138]. As an alternative, circulating apoB can be estimated using an algorithm, but these values are only approximations based on lipid variables such as the total cholesterol, HDL or LDL, and triglycerides, and their clinical relevance was not confirmed [139,140]. Yip et al. provided reference interval values for apoB in children and adolescents [141]. It is usually expressed in mg/dL.

### 3.9. Lp(a)

Lp(a) is a lipoprotein similar to LDL and contains apo(a) and apoB100 in a 1:1 molar ratio [142]. As with other lipoproteins, it acts as a lipid transporter. It is involved in wound healing by binding to fibrin and thus inhibiting fibrinolysis, and transporting cholesterol to injury sites for cell proliferation during tissue repair [143]. Lp(a), similarly to apoB, is also involved in the formation of atherosclerotic plaques. In fact, it causes the activation of inflammatory and prothrombotic processes, and is involved in the formation of atherosclerotic plaque as it increases the proliferation of smooth muscle cells, increases the formation of foamy cells, increases the necrotic nucleus and calcification of atherosclerotic lesions, and upregulates adhesion molecules [144]. In a group of 56,804 participants, Waldeyer et al. showed that elevated Lp(a) conferred an increased risk for major coronary events and cardiovascular disease [145]. Bermudez et al. showed an association between elevated levels of Lp(a) and the onset of MetS [146]; these data were also confirmed by Paredes et al. [147]. Although the influence exerted by Lp(a) in the genesis of MetS was demonstrated by numerous studies, there is very limited evidence for the role of Lp(a) in the pathogenesis of MetS in CCS [19]. Lp(a) can be measured by immunoassay; it is usually expressed in mg/dl, but the correct measurement is in nmol/L [148]. Langer et al. established the upper percentile cut-offs for Lp(a) as follows: ages 3 to 6 months, 14 mg/dL; ages 6.1 to 12 months, 15 mg/dL; ages 1.1 to 9 years, 22 mg/dL; and ages 9.1 to 18 years, 30 mg/dL [149].

## 4. Effectiveness of Biomarkers in Predicting Metabolic Syndrome and Cardiovascular Disease in CCS

In a recent meta-analysis, Pluimakers et al. analyzed the diagnostic and predictive value of MetS-related biomarkers in CCS. They analyzed 175 papers relating to the general population and five studies relating to CCS. They observed that uric acid, adiponectin, hs-CRP, leptin, and apoB can be used as biomarkers in MetS screening of CCS to enhance the early identification of those at high-risk of subsequent complications [19]. They were also able to establish the prognostic value of uric acid and hsCRP in predicting the appearance of MetS. The pooled OR for the association between hyperuricemia and MetS, adjusted for age and sex, was 2.94 (95%CI 2.08–4.15) with an unadjusted pooled OR per unit increase in uric acid of 1.086 (95% CI 1.066–1.106). For hsCRP, they defined an unadjusted pooled AUC of 0.71 (95%CI 0.67–0.74) [19].

Instead, they found no sufficient evidence to confirm the value of candidate biomarkers Lp(a), IL-1, IL-6, and TNF-alpha, although for them, some relevance was shown in the general population [19].

At the moment, no other data are available on the efficacy of biomarkers in the diagnosis of MetS in CCS and we hope future studies will deepen knowledge regarding this subject. The discovery of an early biomarker of MetS will allow identified individuals to undertake lifestyle modifications such as a heart-healthy diet and regular exercise [150].

## 5. Conclusions

MetS is a relevant problem for CCS and is a leading cause of early death. It is currently possible to implement therapeutic strategies and treatments that block the pathogenetic mechanisms of MetS. For this reason, the identification of early biomarkers will greatly improve the survival of CCS. Uric acid and hsCRP are already effective in predicting the occurrence of MetS, and should therefore be included in CCS surveillance protocols and performed at all follow-up evaluations.

However, sufficient data are not yet available for the other biomarkers analyzed in the article, due to the small number of studies available in the scientific literature; future studies may permit a more thorough definition of their efficacy, guaranteeing an improvement in the survival of CCS.

## Figures and Tables

**Figure 1 jpm-12-00880-f001:**
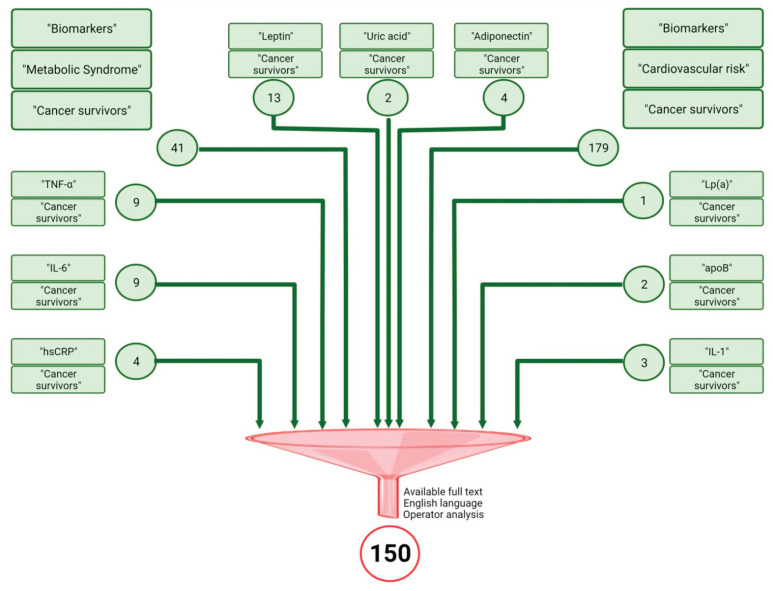
Research methods. The figure shows the results of the research carried out on PubMed. The rectangles contain the search terms. The number of articles obtained from the individual searches are contained in the circles. All articles obtained were analyzed according to the filters noted adjacent to the funnel. The final number of articles included in the review is 149.

**Figure 2 jpm-12-00880-f002:**
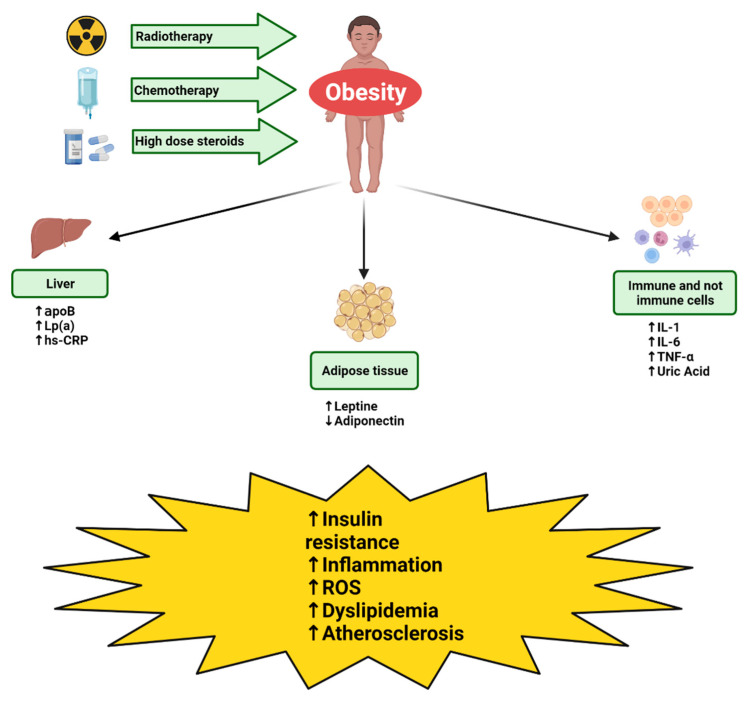
Mechanisms of action of the biomarkers of MetS. Radiotherapy, chemotherapy and high dose steroids cause the appearance of weight gain with a consequent increase in the hepatic production of apoB, Lp(a) and hs-CRP, a reduction in the production of adiponectin by the adipose tissue and a simultaneous reduction in the production of leptin, reduction in the production of IL-1, IL-6, TNF by the immune and non-immune cells, and uric acid. The production of these markers determines insulin resistance, inflammation, dyslipidemia, production of reactive oxygen species (ROS), and atherosclerosis.

**Table 1 jpm-12-00880-t001:** Biomarkers predictive of MetS. The table summarizes the structural characteristics, production sites and mechanisms of action of the biomarkers examined.

Biomarkers	Structure	Production Site	Mechanisms of Involvement in MetS	Changes in MetS
**Adiponectin**	Protein	Adipose tissue Cardiac and skeletal tissue	↑Insulin sensivity ↑Intracellular glucose transport ↑Oxidation of fatty acids ↓ROS and hs-CRP ↓Gluconeogenesis	Reduced
**Leptin**	Protein	Adipose tissue	↓Appetite	Increased (Leptin receptor resistance)
**Uric Acid**	Product of purine metabolism	All cells	↑ROS ↑Inflammatory cytokines ↑Insulin resistance ↑Dyslipidemia	Increased
**hs-CRP**	Protein	Liver cells	↑Inflammation ↑ROS	Increased
**TNF-α**	Cytokine	Immune and non-immune cells	↑Hepatic lipid synthesis ↑Adipose lipolisis ↑Cholesterol biosynthesis ↑Inflammation ↑Vascular insulin resistance	Increased
**IL-1**	Cytokine	Immune and non-immune cells	↑Inflammation ↑Insulin resistance	Increased
**IL-6**	Cytokine	Immune cells Osteoblasts Muscle cells	↑Inflammation ↑Insulin resistance ↑Plasma glucose ↑Free fatty acids	Increased
**Apo B**	Transporter protein	Liver cells	↑Atherosclerosis	Increased
**Lp (a)**	Transporter protein	Liver cells	↑Inflammation ↑Atherosclerosis ↑Vascular muscle cells proliferation ↓Fibrinolisis	Increased

↑ indicates an increase in plasma concentration; ↓ indicates a decrease in plasma concentration.

## Data Availability

Not applicable.

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
