# Peer review of "Biomarkers Predictive of Metabolic Syndrome and Cardiovascular Disease in Childhood Cancer Survivors"

_jpm, 2022, doi:10.3390/jpm12060880_

Round 1

Reviewer 1 Report

Overall, this is an article evaluating a topic that is highly relevant and of great importance.  Clearly this is a very thoroughly researched manuscript.  

Introduction is very strong and concise.  However, there is no methods section.  The authors describe very brief methods as part of the introduction, but don't describe the selection criteria that ultimately led to 147 papers which met inclusion criteria.  A clear methods section, perhaps with a flow chart or diagram would add significant value to the clearly well researched discussion which follows. 

Pathogenesis clearly highlights roles of radiotherapy, steroids and chemotherapy. 

Review of biomarkers is comprehensive and clearly excellently researched. Excellent summary table 1. 

Fleshing out discussion of effectiveness of biomarkers in predicting disease would add significantly to the manuscript.  This would highlight the relevance and provide a clinical context for the important work reviewed in the prior section.  In addition, highlighting the potential interventions and clinical relevance of such interventions as part of this discussion would again add significantly to the manuscript.  A full review of the treatment options is beyond the scope, but this would frame the importance of the findings. 

Some English language / grammar concerns throughout.  Recommend review. 

Author Response

Overall, this is an article evaluating a topic that is highly relevant and of great importance.  Clearly this is a very thoroughly researched manuscript.  

Thank you very much; we have tried to investigate a topic that we think is of great relevance to cancer survivors.

Introduction is very strong and concise.  However, there is no methods section.  The authors describe very brief methods as part of the introduction, but don't describe the selection criteria that ultimately led to 147 papers which met inclusion criteria.  A clear methods section, perhaps with a flow chart or diagram would add significant value to the clearly well researched discussion which follows. 

We have added a paragraph on the methods used and a figure that summarizes them.

Pathogenesis clearly highlights roles of radiotherapy, steroids and chemotherapy. 

Thank you very much.

Review of biomarkers is comprehensive and clearly excellently researched. Excellent summary table 1. 

Thank you very much.

Fleshing out discussion of effectiveness of biomarkers in predicting disease would add significantly to the manuscript.  This would highlight the relevance and provide a clinical context for the important work reviewed in the prior section.  In addition, highlighting the potential interventions and clinical relevance of such interventions as part of this discussion would again add significantly to the manuscript.  A full review of the treatment options is beyond the scope, but this would frame the importance of the findings. 

There are currently no other available articles on the efficacy of biomarkers in the diagnosis of MetS in CCS other than the one mentioned. As suggested by the reviewer, we sought to extend this paragraph by adding our own considerations to help understand the clinical relevance of the topic addressed in the review.

Some English language / grammar concerns throughout.  Recommend review. 

We reviewed English language / grammar concerns.

Reviewer 2 Report

Dear authors,

thank you very much for this interesting manuscript. I have the following comments/ questions:

  1. Introduction: you have a paragraph on your own method there including results - this needs to be reorganized
  2. What is the method of your manuscript? A systematic review? Or on which criteria the articles were selected? Please explain
  3. Normally a manuscript has the following parts: introduction, methods, results and discussion - can you explain your structure in this?
  4. How was your selection on the 9 biomarkers presented?
  5. Under no 4 you have added a short paragraph on the effectiveness of biomarkers and cited a meta-analysis with 175 papers. How much overlap is there between this meta-analysis and your selection?
  6. Conclusion: you are suggesting two biomarkers to measure during surveillance protocols. How often? At what level to intervene?

Author Response

Dear authors,

thank you very much for this interesting manuscript.

Thank you very much; we have tried to investigate a topic that we think is of great relevance to cancer survivors.

I have the following comments/ questions:

Introduction: you have a paragraph on your own method there including results - this needs to be reorganized

We have reorganized the text and inserted a paragraph on methods

What is the method of your manuscript? A systematic review? Or on which criteria the articles were selected? Please explain

In paragraph 1.1 we have included information on the type of review and on the methods used in the bibliographic research

Normally a manuscript has the following parts: introduction, methods, results and discussion - can you explain your structure in this?

Our manuscript is not an original article. It is an integrative review that we have divided into paragraphs whose title summarizes the content.

How was your selection on the 9 biomarkers presented?

As explained in the manuscript, we selected the 9 most frequently analyzed biomarkers in cancer survivors.

Under no 4 you have added a short paragraph on the effectiveness of biomarkers and cited a meta-analysis with 175 papers. How much overlap is there between this meta-analysis and your selection?

Pluimakers et al (Pluimakers, V.G.; van Santen, S.S.; Fiocco, M.; Bakker, M.-C.E.; van der Lelij, A.J.; van den Heuvel-Eibrink, M.M.; Neggers, S.J.C.M.M. Can Biomarkers Be Used to Improve Diagnosis and Prediction of Metabolic Syndrome in Childhood Cancer Survivors? A Systematic Review. Obes. Rev. Off. J. Int. Assoc. Study Obes. 2021, 22, e13312, doi:10.1111/obr.13312.)they have done exceptional work constructing a meta-analysis on the efficacy of new biomarkers in diagnosing metabolic syndrome in cancer survivors. In their work, however, the pathogenetic mechanisms that lead to the production of these biomarkers are not analyzed but only their effectiveness. In our article, we have analyzed the pathogenetic mechanisms of the metabolic syndrome and of the production of these biomarkers in cancer survivors.

Conclusion: you are suggesting two biomarkers to measure during surveillance protocols. How often? At what level to intervene?

In paragraph 5 we specified that they should be executed at all follow-up evaluations.

Round 2

Reviewer 2 Report

Dear authors,

thank you for your revised version. While you have defined now your research method the figure is not clear to me and numbers in the text and legend vs. the figure are not matching (150 vs. 149). This requires clarification.